# Interpretable Surrogate Models: A Clustering Approach for Gaussian Process Posteriors Using Mixed-Integer Quadratic Programming

## Abstract

Gaussian process regression is a flexible Bayesian method for capturing nonlinearity. Although recent advancements allow us to handle various types of tasks by specifying the covariance and likelihood functions, the interpretation of its predictions is sometimes challenging due to the large number of parameters. In this study, we propose a clustering approach to improve the interpretability of Gaussian process posteriors. Assuming that the parameters corresponding to data points within each cluster are identical, the number of parameters in the posterior distribution is reduced. The assignment of data points to clusters is formulated as a mixed-integer quadratic programming problem, with the objective function being a weighted squared error from the mean of the posterior distribution approximated by variational inference. Graph partitioning and decision tree learning can be represented by incorporating linear inequality constraints into this formulation. Experimental results demonstrated that our approach provided significant advantages in enhancing the interpretability of spatial modeling. Moreover, our formulation has produced higher-scoring decision trees compared to Classification and Regression Trees algorithm.

## 1 Introduction

Gaussian process regression is a Bayesian nonlinear regression method (Rasmussen & Williams, 2006). This regression framework can handle various types of tasks by specifying the covariance and likelihood functions. Spatial modeling is a key application of Gaussian process regression (Cressie, 1993). Stationary covariance functions that incorporate the Euclidean metric are widely used to model spatial relationships. Non-Euclidean metrics are specifically designed to capture anisotropic or nonstationary structures (Paciorek & Schervish, 2003; Feragen et al., 2014). Covariance functions can be designed flexibly for tasks beyond geostatistics. There are covariance functions equivalent to an infinitely wide deep neural network (Lee et al., 2018). Likelihood functions based on the behavior of observations significantly impact prediction performance. Specifying non-Gaussian likelihoods that are not conjugate with a Gaussian process prior has emerged as a significant area of research. Variational inference with inducing points is a common approach that maximizes a lower bound on the marginal likelihood using a Gaussian approximation (Quiñonero-Candela & Rasmussen, 2005; Titsias, 2009). Some studies have extended this approach to accommodate general likelihoods (Sheth et al., 2015; Hensman et al., 2015b).

Although Gaussian process regression has flexible expressive power, too many parameters can sometimes make it difficult to interpret the prediction results. Surrogate models trained to approximate the predictions of a black-box model show promise in tackling this challenge (Molnar, 2022). Decision trees are known for being interpretable machine learning models (Quinlan, 1986). They can handle nonlinear regression tasks while maintaining a clear decision-making process. Assuming that the data points in each leaf share a common parameter, a decision tree helps reduce the number of parameters in a Gaussian process posterior distribution. Figure 1 illustrates the behavior of the decision tree surrogate model for a posterior distribution given new inputs. Compared to a decision tree trained directly on observed data, a surrogate model performs better when the distribution of new inputs differs from the training data. This is advantageous for practical applications, such as in marketing and risk management, as it facilitates the creation of groups tailored to target users.

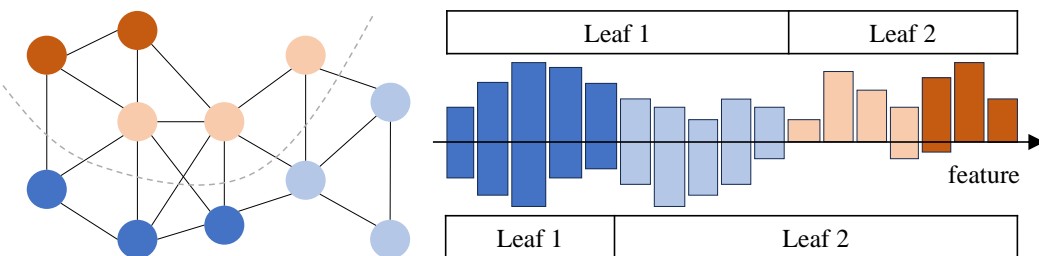

Figure 1: Surrogate models for Gaussian process posteriors. Left: graph partitioning. Right: decision tree learning. Colors illustrate the level of the mean in a posterior. The height from the axis in the right figure represents the distribution of a feature.

Given that a leaf is a type of cluster of data points, clustering with other structures can similarly be used to build surrogate models. Graph partitioning is the process of dividing the nodes of a graph into smaller disjoint subgraphs while minimizing an objective (Nascimento & de Carvalho, 2011; Buluç et al., 2016). This technique is particularly beneficial for clustering data points based on their dependencies in spatial modeling.

In this study, we propose a clustering approach to enhance the interpretability of Gaussian process regression. To train surrogate models for Gaussian process posteriors, it is required to adopt criteria that align with the likelihood functions. We formulate a mixed-integer quadratic programming (MIQP) problem to assign data points to clusters, maximizing the probability density of the posterior distribution approximated through variational inference. The objective function minimizes the weighted squared error from the mean of the Gaussian distribution that approximates the posterior, with weights determined by the inverse of the variance. Incorporating linear inequality constraints into this formulation allows us to represent graph partitioning and decision tree learning. Figure 2 illustrates the overview of our approach. Experiments demonstrate that graph partitioning through the Gaussian process posterior enhances the interpretability of spatial modeling. Moreover, decision tree learning using our formulation has achieved higher scores than Classification and Regression Trees (CART) algorithm (Breiman et al., 1984).

## 2 RELATED WORK

**Local Surrogate Model.** While our approach attempts to approximate the overall behavior of black-box models with interpretable models, other candidate methods are available to explain individual predictions (Guidotti et al., 2018). Local Interpretable Model-agnostic Explanations (LIME) uses local surrogate models trained on perturbed samples generated around a specific instance (Ribeiro et al., 2016). These models approximate the original model's behavior within the local region. Using the concept of SHapley value from cooperative game theory (Shapley, 1953), SHapley Additive exPlanations (SHAP) assigns an importance value to each feature for a particular prediction (Lundberg & Lee, 2017). SHAP values, used as coefficients in a linear function of binary variables corresponding to features, provide valuable properties for enhancing interpretability. Koh & Liang 2017 proposed a method to formalize the impact of a training point on a prediction, providing efficient computation through influence functions derived from robust statistics (Hampel, 1976). This formulation helps us understand the behavior of black-box models during the learning process.

**Non-parametric Clustering.** The probability density function estimated by non-parametric models, such as suport vector machines (Ben-Hur et al., 2002) and Gaussian processes (Kim & Lee, 2007), enables us to indetify high-density regions. The valleys in this probability distribution are interpreted as the boundaries between clusters. Unlike our approach, this clustering method focuses on identifying regions rather than grouping individual data points into clusters and does not aim to preserve the prediction accuracy of the original models.

**Optimal Tree.** An optimal tree is a decision tree constructed to minimize a certain objective function. Although optimal trees may offer both simplicity and precision, building them is an NP-complete task (Hyafil & Rivest, 1976). For this reason, practical algorithms are typically heuristic. The CART algorithm is one of the most widely used heuristics for decision tree learning. Re-

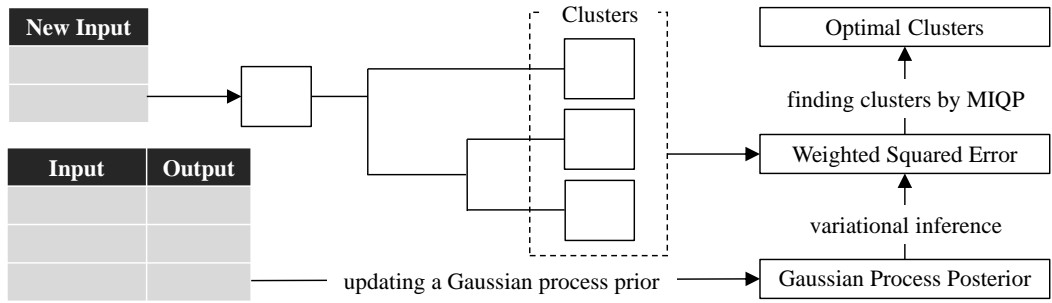

Figure 2: Overview of our approach. To reduce the number of parameters in the Gaussian process posterior distribution, we assign new data points to clusters. The assignment of data points is formulated as an MIQP problem that minimizes the weighted squared error. Graph partitioning and decision tree learning are specific cases of this clustering approach. The figure represents the process of decision tree learning.

cent improvements in computing performance have motivated the development of exact algorithms. Bertsimas & Dunn 2017 proposed a mixed-integer linear programming (MIP) approach to find optimal trees. Since its publication, many subsequent approaches (Hu et al., 2019; Verwer & Zhang, 2019; Aglin et al., 2020; Verhaeghe et al., 2020; Günlük et al., 2021; Demirovic & Stuckey, 2021; Demirović et al., 2022; Zhang et al., 2023) have been proposed. Appendix B shows whether these approaches are classification or regression. For several reasons, building optimal classification trees is easier. Specifically, the binarization technique proposed by (Verwer & Zhang, 2019) enables the efficient conversion of continuous variables into binary ones for classification tasks. While continuous variables can be directly represented by linear inequalities in an MIP formulation, most metrics used in regression tasks do not take an affine form. Our approach aims to minimize the weighted squared error involving continuous variables. To the best of our knowledge, no existing work in the context of optimal trees has yet satisfied this requirement.

## 3 PRELIMINARY

To set the foundation for our approach, we describe Gaussian process regression, MIQP, and two potential surrogate models. Appendix A indicates the list of symbols. Let the domain of inputs denote $\mathcal{X} \equiv \mathcal{X}_1 \times \cdots \times \mathcal{X}_d \subset \mathbb{R}^d$.

### 3.1 GAUSSIAN PROCESS REGRESSION

A Gaussian process $f \sim \mathcal{GP}(\tau(\cdot), k(\cdot, \cdot))$ is a distribution over functions characterized by a mean function $\tau : \mathcal{X} \to \mathbb{R}$ and a covariance function $k : \mathcal{X} \times \mathcal{X} \to (0, \infty)$. For simplicity, we take $\tau(\cdot) = 0$. A stochastic process $\{f(\boldsymbol{x}) \mid \boldsymbol{x} \in \mathcal{X}\}$ is a Gaussian process if and only if the random variables $\{f(\boldsymbol{x}) \mid \boldsymbol{x} \in \mathcal{X}'\}$ for any finite set $\mathcal{X}' \subseteq \mathcal{X}$ follow a multivariate normal distribution. A more detailed introduction is presented in (Rasmussen & Williams, 2006).

**Variational Inference.** Inducing points $\boldsymbol{Z} \equiv (\boldsymbol{z}_i)_{i=1}^m$ with $\boldsymbol{z}_i \in \mathcal{X}$ are used in variational inference for Gaussian process regression. The posterior distribution of $\boldsymbol{u} \equiv (f(\boldsymbol{z}_i))_{i=1}^m$ is approximated by $q(\boldsymbol{u}) \equiv \mathcal{N}(\boldsymbol{u}; \boldsymbol{b}, \boldsymbol{S})$, where $\boldsymbol{b}$ is an $m$-dimensional real vector and $\boldsymbol{S}$ is an $m \times m$ positive-definite matrix. These parameters $(\boldsymbol{Z}, \boldsymbol{b}, \boldsymbol{S})$ are learned to maximize the evidence lower bound given observed data. Let the $(i, j)$-th element of the Gram matrix for $(\boldsymbol{A}, \boldsymbol{B})$ be the return value of a covariance function when the inputs are the $i$-th row of $\boldsymbol{A}$ and the $j$-th row of $\boldsymbol{B}$. We assume that symmetric Gram matrices are positive-definite. For new inpus $\boldsymbol{X} \equiv (\boldsymbol{x}_i)_{i=1}^n$ with $\boldsymbol{x}_i \in \mathcal{X}$, the posterior distribution of $\boldsymbol{f} \equiv (f(\boldsymbol{x}_i))_{i=1}^n$ is approximated by

$$q(\boldsymbol{f}) \equiv \mathcal{N}(\boldsymbol{f}; \boldsymbol{\mu}, \boldsymbol{\Sigma}), \tag{1}$$

where $\boldsymbol{\mu} \equiv \boldsymbol{K}_{\mathbf{uf}}^\top \boldsymbol{K}_{\mathbf{uu}}^{-1} \boldsymbol{b}$, $\boldsymbol{\Sigma} \equiv \boldsymbol{K}_{\mathbf{ff}} + \boldsymbol{K}_{\mathbf{uf}}^\top \boldsymbol{K}_{\mathbf{uu}}^{-1} (\boldsymbol{S} - \boldsymbol{K}_{\mathbf{uu}}) \boldsymbol{K}_{\mathbf{uu}}^{-1} \boldsymbol{K}_{\mathbf{uf}}$, $\boldsymbol{K}_{\mathbf{uu}}$ is the Gram matrix of $(\boldsymbol{Z}, \boldsymbol{Z})$, $\boldsymbol{K}_{\mathbf{uf}}$ is the Gram matrix of $(\boldsymbol{Z}, \boldsymbol{X})$, $\boldsymbol{K}_{\mathbf{ff}}$ is the Gram matrix of $(\boldsymbol{X}, \boldsymbol{X})$, and $\mathcal{N}$ is the probability density function of Gaussian distribution.

## 3.2 MIXED-INTEGER QUADRATIC PROGRAMMING

Minimizing $\frac{1}{2}\boldsymbol{x}^\top \boldsymbol{Q}\boldsymbol{x} + \boldsymbol{c}^\top \boldsymbol{x}$ with respect to an $I$-dimensional real vector $\boldsymbol{x}$ is refered to as an unconstrained quadratic programming problem, where $\boldsymbol{Q}$ is an $I \times I$ real symmetric matrix and $\boldsymbol{c}$ is an $I$-dimensional real vector. If the matrix $\boldsymbol{Q}$ is positive-definite, then $-\boldsymbol{Q}^{-1}\boldsymbol{c}$ is the solution of unconstrained quadratic programming (Boyd & Vandenberghe, 2004). An MIQP problem is formulated by incorporating linear inequality constraints and requiring certain elements of $\boldsymbol{x}$ to be either $0$ or $1$. Algorithms to solve it benefit from the positive-definiteness of $\boldsymbol{Q}$. We refer to this type of MIQP as a positive-definite MIQP.

## 3.3 SURROGATE MODELS

**Directed Acyclic Graph.** A graph is defined by a set of vertices and edges. A directed acyclic graph (DAG) is a graph with directed edges and no cycles. The problem of learning DAGs is central to many areas of machine learning, such as Bayesian network structure learning (Kitson et al., 2023). A connected graph, which is an undirected graph where a path exists between every pair of vertices, represents the connectivity among data points. If a DAG has exactly one leaf (i.e., a node with no child), then it becomes a connected graph when the directions of the edges are ignored. Let a topological ordering $\prec$ denote a binary relation between any two of vertices such that

$$V_i \prec V_j \wedge V_j \prec V_k \Rightarrow V_i \prec V_k \text{ and } V_i \prec V_j \Rightarrow V_i \notin \Pi_j, \tag{2}$$

where $V_i$ is the $i$-th vertex, and $\Pi_i$ is the set of vertices with edges directed towards $V_i$. If and only if a graph is a DAG, a topological ordering exists (Cormen et al., 1990).

**Decision Tree.** Decision trees graphically represent decision-making, illustrating how decisions lead to consequences. A decision tree is a graphical representation of decision-making, illustrating how decisions lead to consequences. A comprehensive description of decision trees can be found in (Quinlan, 1986; Breiman et al., 1984). A path from root to leaf in the tree structure corresponds to a conjunctive rule, and the entire decision tree captures a set of disjunctive rules. Starting from the root node, branches are recursively built down according to the possible values of input features. Each node corresponds to a subset of the input domain. For uniformity of notation, we let the initial domain of the $i$-th feature at the root node be $[\min \mathcal{X}_i, \max \mathcal{X}_i + \epsilon_i)$, where $\epsilon_i$ is a positive real number smaller than the minimum non-zero interval in the $i$-th feature of the inputs. We denote the locations of the two nodes directly below the node located at $o$ as $o \to \text{left}$ and $o \to \text{right}$. Let $[s_{io}, t_{io})$ denote the domain of the $i$-th feature corresponding to the node located at $o$. We adopt one feature at each branch. Then the relation of domains is

$$\begin{cases} s_{io} = s_{io\to\text{left}}, \ t_{io} = t_{io\to\text{right}}, \ t_{io\to\text{left}} = s_{io\to\text{right}} & (\text{if the } i\text{-th feature is adopted}), \\ s_{io} = s_{io\to\text{left}} = s_{io\to\text{right}}, \ t_{io} = t_{io\to\text{left}} = t_{io\to\text{right}} & (\text{otherwise}). \end{cases} \tag{3}$$

Decision trees are commonly used for supervised learning tasks. Given complete data, branches are optimized to maximize a specific metric. Common measures (e.g., information gain, Gini index, and mean squared error) can be represented as a linear sum of local scores computed at the leaves of a tree. Many algorithms rely on these linear metrics, which enable the separate optimization of subtrees within the overall tree.

## 4 APPROACH

In this section, we introduce surrogate models to improve the interpretability of Gaussian process posteriors. Our approach relies on the concept of clustering through a Gaussian process posterior. Graph partitioning and decision tree learning can be modeled by finding clusters under certain constraints. These tasks are formulated as MIQP problems.

### 4.1 CLUSTERING FOR GAUSSIAN PROCESS POSTERIOR

Each data point has a parameter in the framework of Gaussian process regression. While this allows us to deal with various observations flexibly, too many parameters generally pose a disadvantage for the interpretation of estimation results. To improve the explainability of a Gaussian process posterior, we assign new data points to $l$ clusters where the data points in each cluster have a common

parameter. Let $\boldsymbol{\omega} \equiv (\omega_i)_{i=1}^n$ with $\omega_i \in \{1, \cdots, l\}$ denote the data point assignment, and $\boldsymbol{v} \equiv (v_i)_{i=1}^l$ with $v_i \in \mathbb{R}$ denote the parameters corresponding to clusters. From eq. (1), the posterior probability of $\boldsymbol{v}$ and $\boldsymbol{\omega}$ can be approximated by

$$L(\boldsymbol{\omega}, \boldsymbol{v}) \equiv -\frac{n}{2}\log(2\pi) - \frac{1}{2}\log|\boldsymbol{\Sigma}| - \frac{1}{2}(\boldsymbol{W}\boldsymbol{v} - \boldsymbol{\mu})^\top \boldsymbol{\Sigma}^{-1}(\boldsymbol{W}\boldsymbol{v} - \boldsymbol{\mu}), \tag{4}$$

where $\boldsymbol{W}$ is an $n \times l$ matrix satisfying $[\boldsymbol{W}]_{ij} = 1$ if $\omega_i = j$; $[\boldsymbol{W}]_{ij} = 0$ otherwise. For simplicity, we assume that each column in $\boldsymbol{W}$ contains at least one non-zero component. Here we aim to find clusters that maximize $L(\boldsymbol{\omega}, \boldsymbol{v})$. The following lemma is used in this maximization. See appendix C.

**Lemma 4.1.** $\boldsymbol{W}^\top \boldsymbol{\Sigma}^{-1} \boldsymbol{W}$ *is positive-definite.*

From lemma 4.1, the unique mode of $\boldsymbol{v}$ given $\boldsymbol{\omega}$ is $\hat{v}(\boldsymbol{\omega}) \equiv (\boldsymbol{W}^\top \boldsymbol{\Sigma}^{-1} \boldsymbol{W})^{-1} \boldsymbol{W}^\top \boldsymbol{\Sigma}^{-1} \boldsymbol{\mu}$. A naive approach to addressing this optimization is to iterate between optimizing $\boldsymbol{\omega}$ given $\boldsymbol{v}$ and updating $\boldsymbol{v}$ as $\hat{v}(\boldsymbol{\omega})$. See appendix D. While this iteration always converges, the convergence destination depends on the initial state of $\boldsymbol{\omega}$. Therefore, we attempt to maximize $L(\boldsymbol{\omega}, \boldsymbol{v})$ directly. To present the assignment $\boldsymbol{\omega}$, we use binary variables $\boldsymbol{w} = ((w_{ij})_{j=1}^l)_{i=1}^n$ with $w_{ij} \in \{0, 1\}$. Each element $w_{ij}$ corresponds to $[\boldsymbol{W}]_{ij}$. The optimization problem involving $\boldsymbol{w}$ and $\boldsymbol{v}$ is formulated as follows:

$$\text{minimize } (\bar{\boldsymbol{W}}\boldsymbol{v} - \boldsymbol{\mu})^\top \boldsymbol{\Sigma}^{-1}(\bar{\boldsymbol{W}}\boldsymbol{v} - \boldsymbol{\mu}) \text{ subject to } w_{i1} + \cdots + w_{il} = 1 \text{ for all } 1 \leq i \leq n, \tag{5}$$

where $\bar{\boldsymbol{W}}$ is an $n \times l$ matrix satisfying $[\bar{\boldsymbol{W}}]_{ij} = w_{ij}$. To transform the fourth-degree polynomials into quadratic ones, we replace $w_{ij}v_j$ with $\bar{v}_{ij}$ under the following constraint:

$$-Mw_{ij} \leq \bar{v}_{ij} \leq Mw_{ij} \text{ and } v_j - M(1 - w_{ij}) \leq \bar{v}_{ij} \leq v_j + M(1 - w_{ij}), \tag{6}$$

where $M$ is a positive real number such that $[-M, M]^l$ includes $\hat{v}(\boldsymbol{\omega})$ for any $\boldsymbol{\omega}$. Lastly, we introduce the following theorem. See appendix E.

**Theorem 4.2.** *Equation* (5) *can be reformulated as a positive-definite MIQP problem.*

By discarding the covariance among new inputs, our approach becomes equivalent to a regression model trained on the complete data $(\boldsymbol{X}, \boldsymbol{\mu})$ with a weighted squared error. Intuitively, the weights of data points with high uncertainty are small. Consider marginalizing $\boldsymbol{v}$ in $L(\boldsymbol{\omega}, \boldsymbol{v})$. This metric is

$$L(\boldsymbol{\omega}, \hat{v}(\boldsymbol{\omega})) - \frac{1}{2}\log|\boldsymbol{W}^\top \boldsymbol{\Sigma}^{-1} \boldsymbol{W}| + \frac{l}{2}\log(2\pi). \tag{7}$$

See appendix F. The second term indicates the difference from the objective function in the MIQP problem. Although this term restricts the increase in the number of clusters, it cannot be expressed in quadratic form. To leverage efficient algorithms for MIQP formulation, we focus on the maximization of $L(\boldsymbol{\omega}, \boldsymbol{v})$ in this study.

The above discussion lays the foundation to evaluate the approximation performance for Gaussian process posteriors. However, the enhancement of interpretability remains outside the scope of this evaluation. A cluster that is too small often hinders interpretability, even with high predictive performance. Small clusters can also hinder the practicality of clustering. For instance, the profitability of small clusters may be poor when setting advertisements in each cluster. The following constraint ensures that the $i$-th non-empty cluster contains at least $n_0$ data points:

$$n_0 \alpha_i \leq w_{1i} + \cdots + w_{ni} \leq n\alpha_i, \tag{8}$$

where $\alpha_i \in \{0, 1\}$ corresponds to whether the $i$-th cluster is empty ($\alpha_i = 0$) or not ($\alpha_i = 1$). The number of clusters is optimized by incorporating $\alpha_1, \cdots, \alpha_l$ into the variable set of the MIQP problem. Additionally, we often assume that data points within a certain granularity are included in the same cluster. Practical usage constrains the minimum granularity of clusters, such as by city, grade, or age group. The granularity constraint is closely related to ensuring the fairness of machine learning models, as well as other societal requirements. Moreover, this assumption is advantageous for scalable tasks as it reduces the number of variables in the MIQP formulation.

## 4.2 FORMULATION FOR GRAPH PARTITIONING

Spatially connected datapoints in each cluster are easier to make the estimation results more visually comprehensible. In particular, such clusters within physical spaces tend to have more meaningful

$$\text{Ordering} \begin{bmatrix} 0 \le r_{12} + r_{23} - r_{13} \le 1 \\ 0 \le r_{12} + r_{24} - r_{14} \le 1 \\ 0 \le r_{13} + r_{34} - r_{14} \le 1 \\ e_{21} \le 1 - r_{12} \\ e_{31} \le 1 - r_{13} \\ e_{41} \le 1 - r_{14} \\ \vdots \end{bmatrix}$$

$$\text{Separation} \begin{bmatrix} -2(1 - e_{21}) \le (w_{21} - w_{11}) + 2(w_{22} - w_{12}) \le 2(1 - e_{21}) \\ -2(1 - e_{31}) \le (w_{31} - w_{11}) + 2(w_{32} - w_{12}) \le 2(1 - e_{31}) \\ -2(1 - e_{41}) \le (w_{41} - w_{11}) + 2(w_{42} - w_{12}) \le 2(1 - e_{41}) \\ \vdots \end{bmatrix}$$

$$\text{Connectivity} \begin{bmatrix} 1 - \beta_1 \le e_{21} + e_{31} + e_{41} \\ \vdots \end{bmatrix}$$

Figure 3: Example of graph partitioning. Edges in the connected graph are either removed or replaced by directed edges. Each DAG with exactly one leaf corresponds to a cluster.

interpretations. Let the $i$-th data point correspond to the $i$-th vertex of an undirected graph, and let the neighboring data points correspond to the vertices connected by edges to it. We consider the graph partitioning problem, which constructs separate connected graphs by removing edges from the undirected graph. Each separate connected graph corresponds to a cluster. Here we introduce the following theorem. See appendix G.

**Theorem 4.3.** *If and only if an undirected graph is a connected graph, there exists at least one DAG that has exactly one leaf, with edges included in the set of directed edges obtained when the edges in the undirected graph are replaced by bidirectional edges.*

From theorem 4.3, we construct at most $l$ DAGs that can be represented by linear inequalities for an MIQP formulation. Figure 3 illustrates this formulation.

**Separation.** The following condition ensures that there exists no directed edge from the $j$-th vertex in one cluster to the $i$-th vertex in a different cluster:

$$-l(1 - e_{ij}) \le \sum_{1 \le k \le l} k(w_{ik} - w_{jk}) \le l(1 - e_{ij}), \tag{9}$$

where $e_{ij} \in \{0, 1\}$ corresponds to whether the directed edge exists ($e_{ij} = 1$) or not ($e_{ij} = 0$). This condition implies that each cluster has its own separate graph.

**Ordering.** The separate graph within a cluster becomes a DAG when eq. (2) holds. The ordering of vertices and the consistency of edges in this equation are captured by

$$0 \le r_{ij} + r_{jk} - r_{ik} \le 1 \text{ and } e_{ji} \le 1 - r_{ij}, \tag{10}$$

where $r_{ij} \in \{0, 1\}$ satisfies $r_{ij} = 0$ if the order of the $j$-th vertex is higher than the $i$-th vertex; otherwise, $r_{ij} = 1$. Let $r_{ij} = 1 - r_{ji}$. This formulation requires $\frac{1}{2}n(n-1)$ binary variables.

**Connectivity.** Let $\mathcal{C}_i$ denote the set of indices of the vertices that are adjacent to the $i$-th vertex. The following constraints ensure that each DAG has exactly one leaf:

$$1 - \beta_i \le \sum_{j \in \mathcal{C}_i} e_{ji} \text{ for all } 1 \le i \le n, \tag{11}$$

where $\beta_i \in \{0, 1\}$ corresponds to whether the $i$-th vertex is a leaf ($\beta_i = 1$) or not ($\beta_i = 0$). We constraint the number of clusters and leaves by adding $\alpha_1 + \cdots + \alpha_l = \beta_1 + \cdots + \beta_n$.

## 4.3 FORMULATION FOR DECISION TREE LEARNING

Graph partitioning improves the interpretability of Gaussian process posteriors by visualizing the relationships among data points in a low-dimensional space. For higher-dimensional data, simple

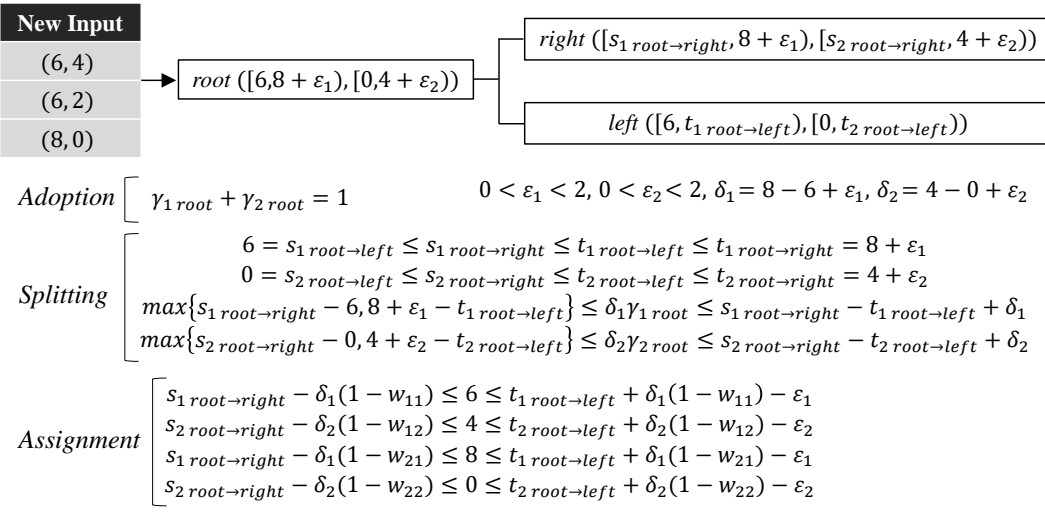

$$\gamma_{1\,root} + \gamma_{2\,root} = 1 \qquad 0 < \varepsilon_1 < 2,\, 0 < \varepsilon_2 < 2,\, \delta_1 = 8 - 6 + \varepsilon_1,\, \delta_2 = 4 - 0 + \varepsilon_2$$

*Splitting*
$$6 = s_{1\,root\to left} \le s_{1\,root\to right} \le t_{1\,root\to left} \le t_{1\,root\to right} = 8 + \varepsilon_1$$
$$0 = s_{2\,root\to left} \le s_{2\,root\to right} \le t_{2\,root\to left} \le t_{2\,root\to right} = 4 + \varepsilon_2$$
$$max\{s_{1\,root\to right} - 6, 8 + \varepsilon_1 - t_{1\,root\to left}\} \le \delta_1 \gamma_{1\,root} \le s_{1\,root\to right} - t_{1\,root\to left} + \delta_1$$
$$max\{s_{2\,root\to right} - 0, 4 + \varepsilon_2 - t_{2\,root\to left}\} \le \delta_2 \gamma_{2\,root} \le s_{2\,root\to right} - t_{2\,root\to left} + \delta_2$$

*Assignment*
$$s_{1\,root\to right} - \delta_1(1 - w_{11}) \le 6 \le t_{1\,root\to left} + \delta_1(1 - w_{11}) - \varepsilon_1$$
$$s_{2\,root\to right} - \delta_2(1 - w_{12}) \le 4 \le t_{2\,root\to left} + \delta_2(1 - w_{12}) - \varepsilon_2$$
$$s_{1\,root\to right} - \delta_1(1 - w_{21}) \le 8 \le t_{1\,root\to left} + \delta_1(1 - w_{21}) - \varepsilon_1$$
$$s_{2\,root\to right} - \delta_2(1 - w_{22}) \le 0 \le t_{2\,root\to left} + \delta_2(1 - w_{22}) - \varepsilon_2$$

Figure 4: Example of decision tree learning. This figure shows a tree with depth 1. Deeper trees can be generated by recursively applying the same split used in this tree.

rules for building clusters can facilitate interpretation. Decision trees offer such rules by creating clusters through orthogonal splits. To learn a decision tree as an MIQP problem, we introduce additional linear inequality constraints to eq. (5). Figure 4 presents this formulation. Aside from this method, constraints for assigning data points can be applied not at each leaf, but at each split. See appendix H. While our approach requires specifying the binary tree structure in advance, eq. (8) allows for empty leaves within the given tree structure. Consequently, the decision tree structure is indirectly optimized.

**Adoption.** The following constraint captures that one feature is adopted at the branch located at $o$:

$$\gamma_{1o} + \cdots + \gamma_{do} = 1, \tag{12}$$

where $\gamma_{io} \in \{0, 1\}$ corresponds to whether the $i$-th feature is adopted ($\gamma_{io} = 1$) or not ($\gamma_{io} = 0$). While this splitting involves a single feature, it can easily be extended to multiple features.

**Splitting.** The common conditions and the case distinctions in eq. (3) are as follows:

$$s_{io} = s_{io\to\text{left}} \le s_{io\to\text{right}} \le t_{io\to\text{left}} \le t_{io\to\text{right}} = t_{io}, \tag{13}$$
$$\max\{s_{io\to\text{right}} - s_{io}, t_{io} - t_{io\to\text{left}}\} \le \delta_i \gamma_{io} \le s_{io\to\text{right}} - t_{io\to\text{left}} + \delta_i, \tag{14}$$

where $\delta_i \equiv \max \mathcal{X}_i - \min \mathcal{X}_i + \epsilon_i$. These constraints are applied sequentially starting from the root node, allowing us to determine the domain associated with each leaf.

**Assignment.** Based on the domains of the leaves, we allocate each data point to a leaf. For the $a$-th leaf located at $o$, the $i$-th data point must satisfy

$$s_{jo} - \delta_j(1 - w_{ia}) \le x_{ij} \le t_{jo} + \delta_j(1 - w_{ia}) - \epsilon_j, \tag{15}$$

where $x_{ij}$ is the $j$-th feature of $\boldsymbol{x}_i$. The number of inequalities for this constraint increases as the number of new inputs grows, making the learning process for decision trees more difficult. The assumption of minimum granularity of clusters helps mitigate this issue, requiring only minor adjustments to the inequalities.

## 5 EXPERIMENT

In this section, we conduct two experiments to evaluate our approach. The first experiment focuses on graph partitioning for geostatistical tasks, with a visualization of the clusters formed on the map. The second addresses decision tree learning as an exact search algorithm, comparing its accuracy with the CART algorithm. The sources of the data and software are indicated in appendix I.2.

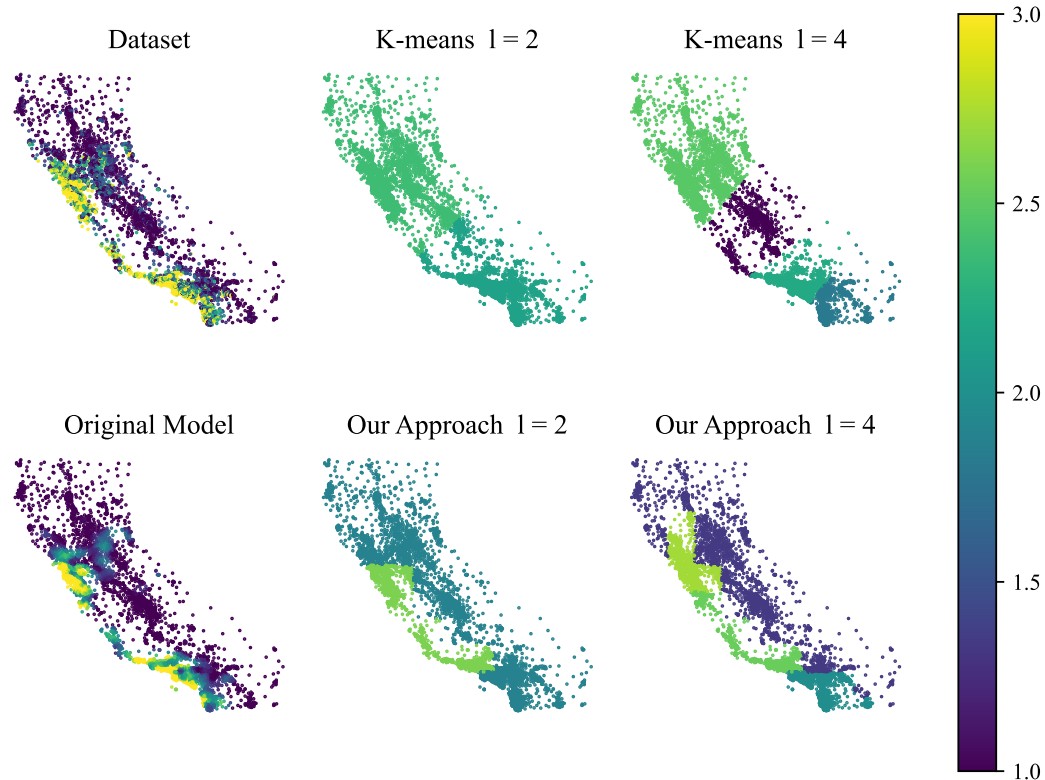

Figure 5: Behavior of graph partitioning. The loss from the k-means algorithm was $0.686$ for $l = 2$ and $0.650$ for $l = 4$. The loss from the our approach was $0.582$ for $l = 2$ and $0.551$ for $l = 4$. For each $l \in \{2, 4\}$, the k-means algorithm converged within a few seconds, and our formulation obtained feasible solutions that were not proven optimal within 5 hours. The color bar represents the observed values or expected values at each data point. The upper bound for the displayed values is 3, and the lower bound is 1.

**Gaussian Process Posteriors.** The posterior distribution was approximated using variational inference methods for Gaussian likelihoods (Hensman et al., 2013) and non-Gaussian likelihoods (Hensman et al., 2015a). The initial $m$ inducing points were obtained using the k-means clustering algorithm (Arthur & Vassilvitskii, 2007). The covariance functions were designed by multiplying a constant kernel with the radius basis function (RBF) kernel and adding white noise. The RBF kernel includes a lengthscale hyperparameter for each dimension of the input space. We present the likelihood function in appendix I.1. The link function was the identity function for the Gaussian, the exponential function for the Poisson, and the probit function for the Bernoulli, respectively. The mean function was $\tau(\cdot) = 0$. We evaluate the approximation performance of surrogate models for this Gaussian process posterior using a loss function defined as the root mean squared error, weighted by the variance of the posterior. Let $n_0 = 1$.

**Computational Environment.** All experiments were conducted on a 64-bit Windows machine with an Intel Xeon W-2265 @ 3.50 GHz and 128 GB of RAM. The code was implemented using Python version 3.7.3. Gurobi Optimizer version 9.1.2 was applied to solve the MIQP problem. Gaussian process regression via variational inference was performed using GPy version 1.12.0. Scikit-learn version 1.5.2 was used for the k-means algorithm. We set all parameters to their default values across all software.

## 5.1 GRAPH PARTIONING

**Setting.** As a spatial dataset, we used location points (i.e., longitude and latitude pairs) as inputs and house prices as outputs in the California Housing dataset (20640 samples). These location

Table 1: Performance of decision tree learning. The mean and standard deviation of the loss in 10-fold cross validation are shown. In all trials, the CART algorithm finished within a few seconds, and the MIQP found feasible solutions that were not proven optimal.

| Name | $m$ | Likelihood | Time Limit | CART | MIQP |
|---|---|---|---|---|---|
| Diabetes | 10 | Gaussian | 1 hour | $10.5 \pm 3.46$ | $9.18 \pm 2.97$ |
| Abalone | 100 | Poisson | 5 hours | $0.0961 \pm 0.00274$ | $0.0932 \pm 0.00362$ |
| Cancer | 10 | Bernoulli | 1 hour | $1.05 \pm 0.124$ | $0.976 \pm 0.0967$ |

points fall within the range $[-124.35, -114.31] \times [32.54, 41.95]$. The regional mesh is expected to help in building interpretable clusters. We assumed that the data points included within each $1 \times 1$ grid belong to a common cluster, and that a data point is connected with another data point if they belong to the same grid or the adjacent grids. A Gaussian process prior was trained on the entire dataset, with $m = 100$ inducing points and a Gaussian likelihood. The new inputs were identical to the inputs used for training. The time limit for our clustering approach in each trial was set to 5 hours. As a baseline, we adopted the k-means clustering algorithm without the regional mesh and the Gaussian process posteriors.

**Result.** As shown in fig. 5, we successfully represented that coastal California housing prices tend to be higher. This result suggests that our approach is capable of capturing clusters with flexible boundaries based on the behavior of the outputs. We believe that existing unsupervised learning methods cannot adequately represent these boundaries. The main drawback of our formulation is the increasing computational cost as the number of clusters grows. For $l = 8$, we were unable to obtain a feasible cluster within the time limit.

## 5.2 DECISION TREE LEARNING

**Setting.** To demonstrate the performance of decision tree learning by the MIQP formulation, we used three dataset: Diabetes (442 samples with $d = 10$), Abalone (4177 samples with $d = 8$), and Cancer (569 samples with $d = 30$). Since our approach cannot handle categorical variables directly, we binarized the column in the Abalone dataset that contains three categories: M, F, and I. We compared the loss from the MIQP formulation with that from the CART algorithm. Ninety percent of each dataset was allocated to obtaining a Gaussian process posterior, while the remaining ten percent was used to build a decision tree surrogate model. The tree structure was set to a depth of 3 with $l = 8$ leaves.

**Result.** Table 1 shows that the MIQP formulation outperformed the CART algorithm in terms of accuracy across the three datasets. However, considering that the MIQP formulation did not find higher accuracy trees for the Abalone dataset within 1 hour from the start, the execution time presents a challenge for larger datasets.

## 6 CONCLUSION

In this study, we proposed a clustering approach to construct surrogate models aimed at enhancing the interpretability of predictions in Gaussian process regression. Graph partitioning and decision tree learning are important tasks for constructing surrogate models. They can be formulated as MIQP problems that maximize a metric to evaluate approximation performance. Experimental results suggest that our approach allows for obtaining flexible clusters in geostatistical tasks. Moreover, decision tree learning using our formulation resulted in higher-scoring trees than those produced by the CART algorithm.

**Limitation.** The objective function in eq. (5) is not a linear metric when the covariance among new inputs is retained. Although our approach can address this challenge, the increasing number of cross terms in the objective function makes finding optimal solutions more difficult. Additionally, even if the covariance is ignored, the computational cost becomes significant with a larger number of new inputs. While the assumption of minimum granularity for clusters helps reduce computational cost, we need to investigate its feasibility for practical tasks.

## ETHICS STATEMENT

This study, which introduces a clustering approach, does not involve any direct ethical concerns. While our approch could be applied in various fields, any potential ethical considerations would depend on the specific context. We should be cautious about fairness or bias in clusters when applying the method in real-world scenarios.

## REPRODUCIBILITY STATEMENT

The proofs of the claims and the supplementary descriptions of the experiments are provided in the appendix. All the source code used in the experiments is included in the supplementary material.

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

## A  NOTATIONS

| Notations | Description |
|---|---|
| $\mathcal{X}$ | $\mathcal{X}_1 \times \cdots \times \mathcal{X}_d \subset \mathbb{R}^d$ |
| $f$ | $f \sim \mathcal{GP}(\tau(\cdot), k(\cdot, \cdot))$ with $\tau: \mathcal{X} \to \mathbb{R}$ and $k: \mathcal{X} \times \mathcal{X} \to (0, \infty)$ |
| $n$ | the number of data points |
| $m$ | the number of inducing points |
| $l$ | the number of clusters |
| $\boldsymbol{X}$ | $(\boldsymbol{x}_i)_{i=1}^n$ with $\boldsymbol{x}_i \in \mathcal{X}$ |
| $\boldsymbol{Z}$ | $(\boldsymbol{z}_i)_{i=1}^m$ with $\boldsymbol{z}_i \in \mathcal{X}$ |
| $\boldsymbol{f}$ | $(f(\boldsymbol{x}_i))_{i=1}^n \in \mathbb{R}^n$ |
| $\boldsymbol{u}$ | $(f(\boldsymbol{z}_i))_{i=1}^m \in \mathbb{R}^m$ with $\boldsymbol{z}_i \in \mathcal{X}$ |
| $\boldsymbol{v}$ | $(v_i)_{i=1}^l$ with $v_i \in \mathbb{R}$ |
| $\boldsymbol{\omega}$ | $(\omega_i)_{i=1}^n$ with $\omega_i \in \{1, \cdots, l\}$ |
| $\boldsymbol{w}$ | $((w_{ij})_{j=1}^l)_{i=1}^n$ with $w_{ij} \in \{0, 1\}$ |
| $q(\boldsymbol{u})$ | $\mathcal{N}(\boldsymbol{u}; \boldsymbol{b}, \boldsymbol{S})$ with an real vector $\boldsymbol{b}$ and an positive-definite matrix $\boldsymbol{S}$ |
| $q(\boldsymbol{f})$ | $\mathcal{N}(\boldsymbol{f}; \boldsymbol{\mu}, \boldsymbol{\Sigma})$ with $\boldsymbol{\mu} = \boldsymbol{K}_{\mathbf{uf}}^\top \boldsymbol{K}_{\mathbf{uu}}^{-1} \boldsymbol{b}$ and $\boldsymbol{\Sigma} = \boldsymbol{K}_{\mathbf{ff}} + \boldsymbol{K}_{\mathbf{uf}}^\top \boldsymbol{K}_{\mathbf{uu}}^{-1}(\boldsymbol{S} - \boldsymbol{K}_{\mathbf{uu}})\boldsymbol{K}_{\mathbf{uu}}^{-1}\boldsymbol{K}_{\mathbf{uf}}$ |
| $\boldsymbol{K}_{\mathbf{ff}}$ | the Gram matrix of $(\boldsymbol{X}, \boldsymbol{X})$ |
| $\boldsymbol{K}_{\mathbf{uf}}$ | the Gram matrix of $(\boldsymbol{Z}, \boldsymbol{X})$ |
| $\boldsymbol{K}_{\mathbf{uu}}$ | the Gram matrix of $(\boldsymbol{Z}, \boldsymbol{Z})$ |
| $\boldsymbol{W}$ | $n \times l$ matrix satisfying $[\boldsymbol{W}]_{ij} = 1$ if $\omega_i = j$; $[\boldsymbol{W}]_{ij} = 0$ otherwise |
| $\bar{\boldsymbol{W}}$ | $n \times l$ matrix satisfying $[\bar{\boldsymbol{W}}]_{ij} = w_{ij} \in \{0, 1\}$ |
| $L(\boldsymbol{\omega}, \boldsymbol{v})$ | $-\frac{n}{2}\log(2\pi) - \frac{1}{2}\log|\boldsymbol{\Sigma}| - \frac{1}{2}(\boldsymbol{W}\boldsymbol{v} - \boldsymbol{\mu})^\top \boldsymbol{\Sigma}^{-1}(\boldsymbol{W}\boldsymbol{v} - \boldsymbol{\mu})$ |
| $\hat{v}(\boldsymbol{\omega})$ | $(\bar{\boldsymbol{W}}^\top \boldsymbol{\Sigma}^{-1}\boldsymbol{W})^{-1}\boldsymbol{W}^\top \boldsymbol{\Sigma}^{-1}\boldsymbol{\mu}$ |
| $[s_{io}, t_{io})$ | the domain of the $i$-th feature located at $o$ |
| $\alpha_i$ | whether the $i$-th cluster is empty ($\alpha_i = 0$) or not ($\alpha_i = 1$) |
| $\beta_i$ | whether the $i$-th vertex is a leaf ($\beta_i = 1$) or not ($\beta_i = 0$) |
| $\gamma_{io}$ | whether the $i$-th feature located at $o$ is adopted ($\gamma_{io} = 1$) or not ($\gamma_{io} = 0$) |
| $\xi_o$ | the split threshold $\xi_o \in (-\infty, \infty)$ for the branch located at $o$ |
| $e_{ij}$ | whether the directed edge $V_j \to V_i$ exists ($e_{ij} = 1$) or not ($e_{ij} = 0$) |
| $r_{ij}$ | $r_{ij} = 0$ if the order of the $j$-th vertex is higher than the $i$-th vertex; otherwise, $r_{ij} = 1$ |
| $\bar{v}_{ij}$ | a binary variable corresponding to $w_{ij}v_j$ |
| $\mathcal{C}_i$ | the set of indices of the vertices that are adjacent to the $i$-th vertex |
| $M$ | a positive real number such that $[-M, M]^l$ includes $\hat{v}(\boldsymbol{\omega})$ for any $\boldsymbol{\omega}$ |
| $\delta_i$ | $\max \mathcal{X}_i - \min \mathcal{X}_i + \epsilon_i$ |
| $\epsilon_i$ | a positive real number smaller than the minimum non-zero interval in the $i$-th feature |
| $n_0$ | the minimum number of data points in a non-empty cluster |
| $\mathcal{N}$ | the probability density function of Gaussian distribution |

## B  OPTIMAL TREES

| Name | Task | Approach |
|---|---|---|
| (Breiman et al., 1984) | classification and regression | heuristic |
| (Bertsimas & Dunn, 2017) | classification | exact |
| (Hu et al., 2019) | classification | exact |
| (Verwer & Zhang, 2019) | classification | exact |
| (Aglin et al., 2020) | classification | exact |
| (Verhaeghe et al., 2020) | classification | exact |
| (Günlük et al., 2021) | classification | exact |
| (Demirovic & Stuckey, 2021) | classification | exact |
| (Demirović et al., 2022) | classification | exact |
| (Zhang et al., 2023) | regression | exact |
| Our Approach | regression | exact |

## C    PROOF OF LEMMA 4.1

For any $n$-dimensional real vector $x \neq 0$, the following holds:

$$
\begin{aligned}
x^\top \Sigma x &= x^\top (K_{\mathbf{ff}} - K_{\mathbf{uf}}^\top K_{\mathbf{uu}}^{-1} K_{\mathbf{uf}}) x + x^\top K_{\mathbf{uf}}^\top K_{\mathbf{uu}}^{-1} S K_{\mathbf{uu}}^{-1} K_{\mathbf{uf}} x \\
&= x^\top K_{\mathbf{uf}}^\top K_{\mathbf{uu}}^{-1} K_{\mathbf{uu}} K_{\mathbf{uu}}^{-1} K_{\mathbf{uf}} x - 2 x^\top K_{\mathbf{uf}}^\top K_{\mathbf{uu}}^{-1} K_{\mathbf{uf}} x + x^\top K_{\mathbf{ff}} x \\
&\quad + (K_{\mathbf{uu}}^{-1} K_{\mathbf{uf}} x)^\top S (K_{\mathbf{uu}}^{-1} K_{\mathbf{uf}} x) \\
&= x_{\mathbf{u}}^\top K_{\mathbf{uu}} x_{\mathbf{u}} + 2 x^\top K_{\mathbf{uf}}^\top x_{\mathbf{u}} + x^\top K_{\mathbf{ff}} x + x_{\mathbf{u}}^\top S x_{\mathbf{u}} \\
&= \begin{bmatrix} x_{\mathbf{u}}^\top & x^\top \end{bmatrix} \begin{bmatrix} K_{\mathbf{uu}} & K_{\mathbf{uf}} \\ K_{\mathbf{uf}}^\top & K_{\mathbf{ff}} \end{bmatrix} \begin{bmatrix} x_{\mathbf{u}} \\ x \end{bmatrix} + x_{\mathbf{u}}^\top S x_{\mathbf{u}}
\end{aligned}
$$

where $x_{\mathbf{u}} \equiv -K_{\mathbf{uu}}^{-1} K_{\mathbf{uf}} x$. From the assumption that symmetric Gram matrices and $S$ are positive-definite, $\Sigma$ is positive-definite. If $x_{\mathbf{v}} \neq 0$, then $W x_{\mathbf{v}} \neq 0$. Therefore, the following holds:

$$
x_{\mathbf{v}}^\top W^\top \Sigma^{-1} W x_{\mathbf{v}} = (W x_{\mathbf{v}})^\top \Sigma^{-1} (W x_{\mathbf{v}}) > 0
$$

for any $l$-dimensional real vector $x_{\mathbf{v}} \neq 0$. From this result, $W^\top \Sigma^{-1} W$ is positive-definite.

## D    ALTERNATING OPTIMIZATION

Here we assume that algorithm 1 does not converge. The following holds:

$$
L(\omega^{(0)}, \hat{v}(\omega^{(0)})) < L(\omega^{(1)}, \hat{v}(\omega^{(0)})) \leq L(\omega^{(1)}, \hat{v}(\omega^{(1)})) < L(\omega^{(2)}, \hat{v}(\omega^{(1)})) \leq \cdots,
$$

where $\omega^{(i)}$ is the state of $\omega$ at the $i$-th itertation in algorithm 1. Therefore, $\omega^{(i)} \neq \omega^{(j)}$ for all $0 \leq i < j$. This is inconsistent with the finite space of $\omega$. Hence, algorithm 1 always converges.

---

**Algorithm 1** Finding Clusters

---

**Input**: $L, \hat{v}$  **Output**: $\omega$

1: **while** Either the evaluation is initial, or $\omega$ has changed since the last evaluation, **do**
2:     $\omega \leftarrow \omega'$ such that $L(\omega', \hat{v}(\omega))$ is maximal under certain constraints.
3: **end while**

---

## E    PROOF OF THEOREM 4.2

The following holds:

$$
\bar{W} v = \begin{bmatrix} w_{11} & \cdots & w_{1l} \\ \vdots & & \vdots \\ w_{n1} & \cdots & w_{nl} \end{bmatrix} \begin{bmatrix} v_1 \\ \vdots \\ v_l \end{bmatrix} = \begin{bmatrix} w_{11} v_1 + \cdots + w_{1l} v_l \\ \vdots \\ w_{n1} v_1 + \cdots + w_{nl} v_l \end{bmatrix}
$$

$$
= \begin{bmatrix} 1 & \cdots & 1 & & & & \\ & & & \ddots & & & \\ & & & & \ddots & & \\ & & & & & 1 & \cdots & 1 \end{bmatrix} \begin{bmatrix} w_{11} v_1 \\ \vdots \\ w_{1l} v_l \\ \vdots \\ w_{n1} v_1 \\ \vdots \\ w_{nl} v_l \end{bmatrix}.
$$

As shown in appendix D, $\Sigma$ is positive-definite. Consequently, using eq. (6), we can get a positive-definite MIQP problem.

# F  MARGINALIZATION

The following holds:

$$L(\boldsymbol{\omega}, \hat{v}(\boldsymbol{\omega})) = -\frac{n}{2}\log(2\pi) - \frac{1}{2}\log|\boldsymbol{\Sigma}| - \frac{1}{2}(\boldsymbol{W}\hat{v}(\boldsymbol{\omega}) - \boldsymbol{\mu})^\top \boldsymbol{\Sigma}^{-1}(\boldsymbol{W}\hat{v}(\boldsymbol{\omega}) - \boldsymbol{\mu})$$

$$= -\frac{n}{2}\log(2\pi) - \frac{1}{2}\log|\boldsymbol{\Sigma}| - \frac{1}{2}\boldsymbol{\mu}^\top \boldsymbol{\Sigma}^{-1}\boldsymbol{\mu} + \frac{1}{2}(\boldsymbol{W}\hat{v}(\boldsymbol{\omega}))^\top \boldsymbol{\Sigma}^{-1}(\boldsymbol{W}\hat{v}(\boldsymbol{\omega}))$$

Using $L(\boldsymbol{\omega}, \hat{v}(\boldsymbol{\omega}))$, the following holds:

$$L(\boldsymbol{\omega}, \boldsymbol{v}) = -\frac{n}{2}\log(2\pi) - \frac{1}{2}\log|\boldsymbol{\Sigma}| - \frac{1}{2}(\boldsymbol{W}\boldsymbol{v} - \boldsymbol{\mu})^\top \boldsymbol{\Sigma}^{-1}(\boldsymbol{W}\boldsymbol{v} - \boldsymbol{\mu})$$

$$= -\frac{1}{2}(\boldsymbol{v} - (\boldsymbol{W}^\top \boldsymbol{\Sigma}^{-1}\boldsymbol{W})^{-1}\boldsymbol{W}^\top \boldsymbol{\Sigma}^{-1}\boldsymbol{\mu})^\top \boldsymbol{W}^\top \boldsymbol{\Sigma}^{-1}\boldsymbol{W}(\boldsymbol{v} - (\boldsymbol{W}^\top \boldsymbol{\Sigma}^{-1}\boldsymbol{W})^{-1}\boldsymbol{W}^\top \boldsymbol{\Sigma}^{-1}\boldsymbol{\mu})$$

$$- \frac{n}{2}\log(2\pi) - \frac{1}{2}\log|\boldsymbol{\Sigma}| - \frac{1}{2}\boldsymbol{\mu}^\top \boldsymbol{\Sigma}^{-1}\boldsymbol{\mu} + \frac{1}{2}(\boldsymbol{W}\hat{v}(\boldsymbol{\omega}))^\top \boldsymbol{\Sigma}^{-1}(\boldsymbol{W}\hat{v}(\boldsymbol{\omega}))$$

$$= \log \mathcal{N}(\boldsymbol{v}; \hat{v}(\boldsymbol{\omega}), (\boldsymbol{W}^\top \boldsymbol{\Sigma}^{-1}\boldsymbol{W})^{-1})$$

$$+ L(\boldsymbol{\omega}, \hat{v}(\boldsymbol{\omega})) - \frac{1}{2}\log|\boldsymbol{W}^\top \boldsymbol{\Sigma}^{-1}\boldsymbol{W}| + \frac{l}{2}\log(2\pi).$$

Therefore, the marginalization of $\boldsymbol{v}$ in $L(\boldsymbol{\omega}, \boldsymbol{v})$ is

$$L(\boldsymbol{\omega}, \hat{v}(\boldsymbol{\omega})) - \frac{1}{2}\log|\boldsymbol{W}^\top \boldsymbol{\Sigma}^{-1}\boldsymbol{W}| + \frac{l}{2}\log(2\pi).$$

# G  PROOF OF THEOREM 4.3

It is evident that no specified DAG exists if the undirected graph is disconnected. We prove that at least one specified DAG exists if the graph is connected. Consider adding vertices to the set of vertices one by one. When adding a vertex, we set a directed edge from the new vertex to each vertex already in the set, provided an undirected edge exists between them. Let the topological ordering correspond to the order in which vertices are added. Since the directed edges satisfy eq. (2), the graph obtained through this procedure is a DAG. The starting vertex in this procedure is a leaf. Given that the undirected graph is connected, we set at least one directed edge when adding each vertex, except for the initial one. Therefore, at least one specified DAG exists if the undirected graph is a connected graph.

# H  ASSIGNMENTS AT BRANCH NODES

We replace the constraints for splitting and assignment in our approach. For the $a$-th leaf that can descend from the branch located at $o$, we consider whether the leaf is on the left or right side of the branch. The $i$-th data point must satisfy

$$\begin{cases} \gamma_{1o}x_{i1} + \cdots + \gamma_{do}x_{id} \le \xi_o + \max_{1 \le j \le d}\delta_j(1 - w_{ia}) - \min_{1 \le j \le d}\epsilon_j & \text{(the left side),} \\ \gamma_{1o}x_{i1} + \cdots + \gamma_{do}x_{id} \ge \xi_o - \max_{1 \le j \le d}\delta_j(1 - w_{ia}) & \text{(the right side),} \end{cases}$$

where $\xi_o \in (-\infty, \infty)$ is the split threshold for the branch located at $o$. This formulation that does not requires to place constraints on each dimension is almost the same as that in (Bertsimas & Dunn, 2017). For higher-dimensional data, the number of constraints in this formulation tends to be smaller than in ours. However, the minimum granularity of clusters cannot be used to reduce the number of inequalities.

# I  EXPERIMENT

## I.1  LIKELIHOOD FUNCTIONS

Let $\Phi$ denote the cumulative distribution function of standard normal distribution, $\Phi_f$ denote $\Phi(f(\boldsymbol{x}))$, and $\phi_f$ denote $\frac{\partial \Phi(f(\boldsymbol{x}))}{\partial f(\boldsymbol{x})}$.

| Function | $p(y \mid f(\boldsymbol{x}))$ | $\frac{\partial \log p(y \mid f(\boldsymbol{x}))}{\partial f(\boldsymbol{x})}$ | $\frac{\partial^2 \log p(y \mid f(\boldsymbol{x}))}{\partial f(\boldsymbol{x})^2}$ |
|---|---|---|---|
| Gaussian | $\frac{1}{\sqrt{2\pi}\sigma} \exp\left(-\frac{(y-f(\boldsymbol{x}))^2}{2\sigma^2}\right)$ | $\frac{y-f(\boldsymbol{x})}{\sigma^2}$ | $-\frac{1}{\sigma^2}$ |
| Poisson | $\frac{\exp(-\exp(f(\boldsymbol{x}))+yf(\boldsymbol{x}))}{y!}$ | $-\exp(f(\boldsymbol{x})) + y$ | $-\exp(f(\boldsymbol{x}))$ |
| Bernoulli | $\Phi_f^y(1-\Phi_f)^{1-y}$ | $\frac{\phi_f}{\Phi_f}y + \frac{-\phi_f}{1-\Phi_f}(1-y)$ | $-\frac{f\phi_f\Phi_f+\phi_f^2}{\Phi_f^2}y - \frac{\phi_f^2-f\phi_f+f\phi_f\Phi_f}{(1-\Phi_f)^2}(1-y)$ |

## I.2 DATA AND SOFTWARE

| Asset | URL |
|---|---|
| GPy | https://www.gurobi.com/ |
| Scikit-learn | https://scikit-learn.org/stable/ |
| California | https://www.dcc.fc.up.pt/˜ltorgo/Regression/cal_housing.html |
| Diabetes | https://www4.stat.ncsu.edu/˜boos/var.select/diabetes.html |
| Abalone | https://archive.ics.uci.edu/ |
| Cancer | https://archive.ics.uci.edu/ |

