# OpenReview forum: "Interpretable Surrogate Models: A Clustering Approach for Gaussian Process Posteriors Using Mixed-Integer Quadratic Programming"
_ICLR.cc/2025/Conference — Submitted to ICLR 2025_

### Official Review · Reviewer_XALY · 2024-10-31

**Soundness:** 2
**Presentation:** 2
**Contribution:** 2
**Rating:** 3
**Confidence:** 2

**Summary:**

This paper explores enhancing interpretability in Gaussian Process (GP) regression by developing a surrogate model that leverages clustering, graph partitioning, and decision trees. By employing this clustering approach, the authors aim to group predictions into more interpretable segments. The model formulation as a mixed-integer quadratic programming (MIQP) problem optimizes a weighted squared error between the predicted values and the mean of the posterior GP distribution, which is approximated using variational inference. This approach seeks to balance the model’s interpretability with prediction accuracy, providing a structured methodology for creating interpretable surrogates in complex GP regression tasks.

**Strengths:**

The paper is well-constructed, with an almost solid contribution to the field. The writing quality is high, and the problem formulation and related work are well-explained, providing a clear foundation for understanding the approach. The MIQP formulation is particularly noteworthy and offers a promising avenue for further development.

**Weaknesses:**

* Initial Definition of the Problem: The problem definition begins with a discussion on the interpretability of Gaussian Processes (GPs), where the first concern arises. GPs are often more interpretable than many machine learning models, particularly due to their probabilistic structure and flexible kernel choices that accommodate domain-specific assumptions. However, as complexity increases (e.g., with higher-dimensional data, and complex kernels), interpretability tends to diminish. The paper could improve its problem definition by clearly specifying which aspect of GP interpretability it seeks to address. For instance, does it aim to handle interpretability in high-dimensional data, manage the interpretability of GPs with complex kernel structures, or focus on non-stationary models? By explicitly defining these directions, the study could clarify the scope and impact of its contributions, helping readers better understand the specific interpretability challenges it addresses.

* Interpretation of GPs with a Large Number of Parameters: A large number of parameters poses challenges in the context of complex kernels and high-dimensional datasets. However, the estimation of these parameters occurs during the training phase, and this paper does not address that step. Consequently, the parameters cannot be altered or modified to enhance interpretability. While the paper identifies the parameters as a source of the interpretability problem, it does not offer any solutions to this issue.

* Novelty and Consistency: If clustering is performed before training, the problem is transformed into a distributed Gaussian process. Predicting new test data points within one or multiple clusters has been explored previously in this field. However, conducting clustering after training and solely on test points is somewhat confusing. In practical scenarios, we do not receive all new points simultaneously; instead, data is entered gradually. How can we perform partitioning under these circumstances?

* Complexity and Computational Cost: Interpretability becomes an issue in Gaussian processes (GPs) as complexity increases. GPs are generally expensive prediction models. However, the authors have integrated this complex method with mixed-integer quadratic programming (MIQP), which is computationally intensive due to its combinatorial nature and the challenges posed by non-linearities in the objective function. It does not appear that this approach can be practically applied, even if it potentially improves interpretability.

* Inadequate Numerical Experiments: The numerical analysis presented in the paper fails to substantiate the main claims. Simply outperforming conventional K-means clustering does not support the key assertions made. Other baselines could have been employed for comparison in the experiments, but they were not utilized in this paper.

**Questions:**

See weaknesses.

---

> ### Author Response · Authors · 2024-11-26
>
> We deeply appreciate your insightful comments, which will greatly enhance the quality of our paper.
> Although a major revision is needed, we will carefully address your feedback in our next submission.
> Thank you once again for your time and effort in reviewing our paper.

---

### Official Review · Reviewer_wRyA · 2024-11-02

**Soundness:** 2
**Presentation:** 1
**Contribution:** 2
**Rating:** 3
**Confidence:** 5

**Summary:**

This paper introduces a clustering approach to improve the interpretability of Gaussian Process (GP) regression. By assuming that parameters within each cluster are identical, it reduces the number of parameters in the GP posterior, making the predictions easier to interpret. The clustering is formulated as a mixed-integer quadratic programming problem, with a weighted squared error objective based on the posterior mean approximated by variational inference. The approach also incorporates graph partitioning and decision tree learning through linear constraints.

**Strengths:**

The idea of combining clustering and GPR seems interesting.

**Weaknesses:**

1. My major concern is about the presentation. Some sections are hard to follow. For example, the first paragraph in the introduction, the relationship between sentences are not clear to me, it's more like a stack of facts.

2. The main goal & contribution is not very clear to me. According to the abstract, the goal is to improve the interpretability of GPR. However, in 5.2, only prediction accuracy was discussed, but the interpretability was completely overlooked.

3. I've been confused by "parameters". What do the authors mean in terms of parameters in the GPR setup?

4. The empirical study can be improved. For example, the clustering results are only compared with k-means, but there are quite a few existing spatial clustering methods that are sometimes better than k-means. Similarly, for 5.2, only CART was considered.

**Questions:**

1. Can the authors explain what does "parameter" mean?

2. Can the authors summarize and highlight the main goal and contribution of this manuscript?

3. More comprehensive experiments are expected.

---

> ### Author Response · Authors · 2024-11-26
>
> We deeply appreciate your insightful comments, which will greatly enhance the quality of our paper.
> Although a major revision is needed, we will carefully address your feedback in our next submission.
> Thank you once again for your time and effort in reviewing our paper.

---

### Official Review · Reviewer_ADqV · 2024-11-03

**Soundness:** 3
**Presentation:** 2
**Contribution:** 2
**Rating:** 5
**Confidence:** 3

**Summary:**

The manuscript proposes computational methods to enhance the interpretability of the Gaussian process (GP) posterior. The methods are based on clustering of the GP posterior mean values, where a single parameter is used to approximate the posterior mean values of the data points in the same cluster and this approximation is formulated as the minimization of the weighted (the weights are derived from the posterior covariance matrix) squared loss using mixed integer quadratic programming (MIQP). The manuscript shows that two surrogate models, graph partitioning and decision tree, can be implemented in the MIQP formulation with additional linear inequality constraints.

**Strengths:**

Although the proposed methods have clear disadvantages (weaknesses), the MIQP formulations for graph partitioning and decision tree learning look novel.

**Weaknesses:**

1. The manuscript seems to have failed to provide attractive applications of the proposed methods to real-world problems. The experiments include only small datasets (even they look like toy datasets). The designed experiments included in the manuscript do not show the significance of the proposed methods.

2. The proposed methods seem to suffer from high computational requirements. It seems that the proposed methods could not handle these small data sets. The manuscript does not provide any computational analysis of the proposed methods. As a result, it is difficult to understand how much computational resources the proposed methods require to solve the given problems.

**Questions:**

1.	I wonder why the authors chose to approximate the GP posterior distribution rather than the GP predictive distribution. In fact, the proposed methods are designed to approximate the GP posterior mean function values with the GP posterior covariance matrix, not the entire GP posterior distribution (i.e., a single (mode) function in the entire function space). In addition, the GP posterior mean and covariance are already approximated ones since the sparse approximation was used instead of the full GP.

2. Please provide some details for the results reported in Table 1.
a. How was the number of inducing points, m, chosen for the data sets?
b. How was the decision tree (CART) trained?
c. It is unclear why model accuracy can be measured by evaluating the values of the loss function (from the MIQP formulation?). In addition, it is unclear whether it is fair to compare the loss function of the two methods, since the proposed methods would provide a solution that minimizes this loss function by design.

---

> ### Author Response · Authors · 2024-11-26
>
> We deeply appreciate your insightful comments, which will greatly enhance the quality of our paper.
> Although a major revision is needed, we will carefully address your feedback in our next submission.
> Thank you once again for your time and effort in reviewing our paper.

---

### Official Review · Reviewer_LWD9 · 2024-11-05

**Soundness:** 3
**Presentation:** 2
**Contribution:** 2
**Rating:** 3
**Confidence:** 4

**Summary:**

This paper proposes using a mixed-integer quadratic programming algorithm to cluster regression coefficients in Gaussian process regression. The approach is further extended to applications in graph partitioning and decision tree growth. While the proposed algorithm is interesting and potentially valuable, I find the paper difficult to read, as it attempts to cover numerous loosely connected topics.

**Strengths:**

Using mixed integer programing as an alternative approach to iterative clustering algorithm such as K-means is indeed an interesting idea and I find the formulation in (5)-(6) quite clever.

**Weaknesses:**

The title of the paper is "...for Gaussian process posteriors," yet the authors delve into topics like graph partitioning and decision tree growth, which don’t seem directly related to Gaussian process regression. This shift from the main theme feels distracting and makes the paper harder to follow. I would have preferred a more focused exploration of Gaussian process regression rather than these loosely connected topics. I also feel that the advantage of the proposed algorithm is not well justified.

**Questions:**

1. In my opinion, the may contribution of the paper should be highlighted as the advantage of using mixed-integer optimization over the iterative clustering algorithm. Indeed, as the authors pointed out, on lines 227-229, the weakness of the iterative clustering algorithm is that it can be trapped in local optimizers. Therefore, the authors should demonstrate why using mixed-integer optimization can overcome such a draw back, either through convergence analysis or extensive simulation studies. But I do not see any of such analysis in the paper, which is a bit disappointing.
2. The authors claim that by grouping the parameters, one can improve the interpretability of the Gaussian process regression coefficients. I fails to see why this is the case. For Gaussian process regression, it the the estimated functions or surfaces that matter most, not the regression coefficients. Please elaborate on the claimed "interpretability". In fact, grouping the coefficients, there is a chance of over-smooth the estimated functions or surfaces if the number of clusters are small. At least some simulation studies should be carried out to investigate these issues.
3. How does the K-means algorithm work in Figure~5? It looks like it is just clustering the spatial locations? Please elaborate.
4. For the decision tree algorithm, it is well-know that a single decision tree is not stable and sub-optimal in capturing the non-linear regression relationship. Ensemble methods  such as random forest and boosting are much better. Could the proposed algorithm scale up to these method computationally?

---

> ### Author Response · Authors · 2024-11-26
>
> We deeply appreciate your insightful comments, which will greatly enhance the quality of our paper.
> Although a major revision is needed, we will carefully address your feedback in our next submission.
> Thank you once again for your time and effort in reviewing our paper.

---

### Meta-Review · Area_Chair_DJPJ · 2024-12-17

**Metareview:**

Although the paper contains some interesting general ideas about GP regression and its interpretability, there are currently way too many open questions and concerns, such as:

- unclear potential advantage of  mixed-integer programming over "classical" iterative clustering: why is is any better on a conceptual level
- unclear precise meaning of the proposed parameter grouping step on overall interpretability of GP regression.
- unclear experiments: why focus exclusively on prediction accuracy, if interpretability seems to be the main motivation?
- unclear advantage of ensemble methods like random forests.
Therefore, I recommend rejection of this paper

**Additional Comments On Reviewer Discussion:**

None of the potential weaknesses of the paper could be addressed in a clear way by the authors in their rebuttal.

---

### Decision · Program_Chairs · 2025-01-22

Reject